# Clinico-Pathological Features, Outcomes and Impacts of COVID-19 Pandemic on Patients with Early-Onset Colorectal Cancer: A Single-Institution Experience

**DOI:** 10.3390/cancers15174242

**Published:** 2023-08-24

**Authors:** Daniel Martinez-Perez, David Viñal, Jesús Peña-Lopez, Diego Jimenez-Bou, Iciar Ruiz-Gutierrez, Sergio Martinez-Recio, María Alameda-Guijarro, Antonio Rueda-Lara, Gema Martin-Montalvo, Ismael Ghanem, Ana Belén Custodio, Lucia Trilla-Fuertes, Angelo Gamez-Pozo, Antonio Barbachano, Javier Rodriguez-Cobos, Pilar Bustamante-Madrid, Asuncion Fernandez-Barral, Aurora Burgos, Maria Isabel Prieto-Nieto, Laura Guerra Pastrian, José Manuel González-Sancho, Alberto Muñoz, Jaime Feliu, Nuria Rodríguez-Salas

**Affiliations:** 1Department of Medical Oncology, Central University Hospital of Asturias, 33011 Oviedo, Spain; dmartinezperez@salud.madrid.org (D.M.-P.); jaime.feliu@salud.madrid.org (J.F.); nuria.rodriguez@salud.madrid.org (N.R.-S.); 2Department of Medical Oncology, Hospital Universitario La Paz, 28046 Madrid, Spain; jpenal@salud.madrid.org (J.P.-L.); djimenezb@salud.madrid.org (D.J.-B.); iciar.ruiz@salud.madrid.org (I.R.-G.); malamedag@salud.madrid.org (M.A.-G.); aruedal@salud.madrid.org (A.R.-L.); gema.martinmontalvo@salud.madrid.org (G.M.-M.); ismael.ghanem@salud.madrid.org (I.G.); ana.custodio@salud.madrid.org (A.B.C.); 3Department of Medical Oncology, Hospital de la Santa Creu i Sant Pau, 08025 Barcelona, Spain; smartinezre@santpau.cat; 4Instituto de Investigación Sanitaria del Hospital Universitario La Paz (IdiPAZ), 28046 Madrid, Spain; abarbachano@iib.uam.es (A.B.); jrcobos@iib.uam.es (J.R.-C.); pbustamante@iib.uam.es (P.B.-M.); afbarral@iib.uam.es (A.F.-B.); iprieto@intermic.com (M.I.P.-N.); josemanuel.gonzalez@uam.es (J.M.G.-S.); amunoz@iib.uam.es (A.M.); 5Molecular Oncology and Pathology Lab, Institute of Medical and Molecular Genetics-INGEMM, Hospital Universitario La Paz-IdiPAZ, 28046 Madrid, Spain; lucia.trilla@salud.madrid.org (L.T.-F.); angelo.gamez@salud.madrid.org (A.G.-P.); 6Instituto de Investigaciones Biomédicas Alberto Sols, Consejo Superior de Investigaciones Científicas (CSIC), Universidad Autónoma de Madrid (UAM), 28049 Madrid, Spain; 7Centro de Investigación Biomédica en Red de Cáncer (CIBERONC), 28029 Madrid, Spain; laura.guerra@salud.madrid.org; 8Department of Gastroenterology, Hospital Universitario La Paz, 28046 Madrid, Spain; aurora.burgos@salud.madrid.org; 9Department of Surgery, Hospital Universitario La Paz, 28046 Madrid, Spain; 10Department of Pathology, Hospital Universitario La Paz, 28046 Madrid, Spain; 11Catedra UAM-AMGEN, 28046 Madrid, Spain

**Keywords:** early-onset colorectal cancer, COVID-19 pandemic, prognosis

## Abstract

**Simple Summary:**

The rising incidence of colorectal cancer (CRC) among young patients (≤50 years) is alarming. We included all patients with pathologically confirmed diagnoses of CRC at Hospital Universitario La Paz from October 2016 to December 2021. A total of 1475 patients diagnosed with CRC were included, eighty (5.4%) of whom had EOCRC. Aggressive pathological features, such as T, N stage and metastatic presentation at diagnosis; perineural invasion; tumor budding; high-grade tumors; and signet ring cell histology, were higher in the early-onset group. Patients with metastatic EOCRC HAD a significantly longer median OS than the older cohort. Regarding COVID-19 pandemic, more patients with COVID-19 were diagnosed with metastatic disease (61%) after the lockdown. The long-term consequences of COVID-19 are yet to be determined.

**Abstract:**

Background: The rising incidence of colorectal cancer (CRC) among young patients is alarming. We aim to characterize the clinico-pathological features and outcomes of patients with early-onset CRC (EOCRC), as well as the impacts of COVID-19 pandemic. Methods: We included all patients with pathologically confirmed diagnoses of CRC at Hospital Universitario La Paz from October 2016 to December 2021. The EOCRC cut-off age was 50 years old. Results: A total of 1475 patients diagnosed with CRC were included, eighty (5.4%) of whom had EOCRC. Significant differences were found between EOCRC and later-onset patients regarding T, N stage and metastatic presentation at diagnosis; perineural invasion; tumor budding; high-grade tumors; and signet ring cell histology, with all issues having higher prevalence in the early-onset group. More EOCRC patients had the RAS/ BRAF wild type. Chemotherapy was administered more frequently to patients with EOCRC. In the metastatic setting, the EOCRC group presented a significantly longer median OS. Regarding the COVID-19 pandemic, more patients with COVID-19 were diagnosed with metastatic disease (61%) in the year after the lockdown (14 March 2020) than in the pre-pandemic EOCRC group (29%). Conclusions: EOCRC is diagnosed at a more advanced stage and with worse survival features in localized patients. More patients with EOCRC were diagnosed with metastatic disease in the year after the COVID-19 pandemic lockdown. The long-term consequences of COVID-19 are yet to be determined.

## 1. Introduction

Colorectal cancer (CRC) is the third most common cancer worldwide and the second most common cause of cancer-related death, with an estimated 1.9 million new diagnoses (accounting for 10% of all cancer diagnosis) and 935,000 deaths in 2020 [1]. Its incidence is approximately four times higher in developed countries than in developing countries, with the highest rates found in Europe, Australia, and North America [1].

Early-onset colorectal cancer (EOCRC) is defined as adults aged <50 years at the time of CRC diagnosis [2]. The median age of diagnosis of CRC was 67 years old in the USA in the period 2013–2017; 68% of patients were 65 years old or older, and 12% of patients were 50 years old or younger [3]. Increases in EOCRC incidence by 1–4% annually in a 10-year period in several high-income countries (HIC) are worrying (Canada by 3.4%, New Zealand by 2.9%, Australia by 2.6% and the U.K. by 1.4%), especially given decreases in incidences of later-onset CRC recorded in those same countries (Canada by 1.9%, New Zealand by 3.4%, Australia by 1.6% and the U.K. by 1.2%) [4,5]. In this context, a microsimulation analysis reported that the efficiency ratio (ER) of lowering the screening initiation for CRC at 45 years old was 32, which is well below the threshold of 39 that the US Preventive Services Task Force (USPSTF) considers to represent an efficient intervention [6]. This evidence has led to the USPSTF recommending screening initiation for CRC at 45 years old [7].

The reasons for the rising incidence of EOCRC are not well understood. The rise is more prominent in the developed world, with a particularly large impact on non-Hispanic White people [2]. Evidence suggests that changes in dietary patterns and lifestyle factors in HIC, such as animal-source foods, antibiotic use, and sedentary lifestyles, are contributing to this increasing incidence of EOCRC [3,8]. Other possible risk factors for EOCRC include male sex, a family history of CRC, hyperlipidemia, high intake of processed meats, high alcohol consumption or inflammatory bowel disease [9,10,11,12]. On the contrary, dietary factors, such as higher intake of vegetables, vitamins B9, C and E, beta-carotenes and fish, have been associated with lower risk of EOCRC [6,13].

Several clinico-pathological features are more common among patients with EOCRC, including the location of the primary tumor in the left colon [14,15], higher prevalence of poor tumor cell differentiation, signet-ring cells, and high microsatellite instability (MSI-H) due to germline mutations in the DNA mismatch repair [3]. Tumors are usually diagnosed at a more advanced stage [16,17,18] due, to some extent, to the lower suspicion of cancer in this age group. In fact, patients with EOCRC have a longer duration of symptoms at presentation and a longer delay in the time until diagnosis than older patients. This delay has also been problematic during the COVID-19 pandemic [3]. The fear of being infected with SARS-CoV2 and the collapse of healthcare systems, including the primary care and emergency departments, has led to further delays in cancer diagnosis. The real impacts of long-term consequences of the COVID-19 pandemic on vulnerable subgroups of patients, such as EOCRC, are yet to be determined.

We aim to characterize the clinico-pathological features and outcomes of patients with EOCRC diagnosed in our hospital area, as well as analyze the impacts of the COVID-19 pandemic on the diagnosis and initial staging of patients with EOCRC.

## 2. Materials and Methods

This study was a single-institution retrospective observational study. We included all patients with histologically confirmed CRC between October 2016 and December 2021 at La Paz University Hospital, Madrid (Spain). This study was approved by the Ethics Committee of La Paz University Hospital and conducted in accordance with ethical standards of the Helsinki Declaration of the World Medical Association. Baseline disease, demographics, clinical data, pathological and molecular data and treatment characteristics were analyzed using the medical records of each patient. All data were presented as mean +/− SD. Differences between groups were evaluated via independent t tests for continuous variables and χ^2^ tests for categorical variables. Yates’ correction was applied when necessary.

Disease-free survival (DFS) was calculated from the date of the surgery to the date of tumor recurrence or death or the last follow-up. Overall Survival (OS) was defined as the time between the date of diagnosis and the date of death or last follow-up. The analysis was performed up to a data cut-off date of 30 October 2022. The relationships between DFS and OS and each of the variables were analyzed using the log-rank test. Survival analysis was performed using the Kaplan–Meier method. Univariate Cox regression analyses and the multivariate proportional hazards regression model were used to identify independent prognostic factors. In the multivariate analysis, we included the variables significantly associated with DFS and OS in the univariate analysis, as well as other clinically relevant variables. All statistical analyses were carried out using SPSS v.25.

To study the impacts of COVID-19 on patients with EOCRC, we established two cohorts and compared the main characteristics at presentation between the cohort diagnosed before the initial date of the lockdown in Spain due to the COVID-19 pandemic, i.e., 14th March 2020 (pre-pandemic group), and the cohort diagnosed during the 365 days after that date (post-pandemic group).

## 3. Results

A total of 1475 patients were diagnosed with CRC, eighty (5.4%) of whom were <50 years old. Baseline characteristics are depicted in Table 1. Patients with EOCRC were more frequently diagnosed with metastatic disease (34% vs. 21%; *p* = 0.005), T4 cancer stage (44% vs. 31%; *p* = 0.05) and N positive disease (69% vs. 47%; *p* = 0.001). More patients in the EOCRC group compared to the older group had characteristics associated with worse prognosis, such as perineural invasion (34% vs. 22%; *p* = 0.029), signet ring cell (7% vs. 2%, respectively; *p* = 0.02) and high-grade tumors (19% vs. 8%, *p* = 0.004). The prevalence of dMMR/MSI-high tumors was higher among patients with EOCRC (14% vs. 9%), although the difference was not statistically significant. Mutational status was assessed in 422 patients, finding a higher percentage of wild-type (RAS/BRAF non-mutant) tumors in EOCRC than in the older cohort (62% vs. 34%; *p* = 0.02). Regarding treatment received, no differences were observed in the number of patients who underwent surgery (81% vs. 84%, respectively; *p* = 0.490). More patients with EOCRC received chemotherapy (85% vs. 62%; *p* < 0.001). In patients with localized disease (*n* = 1161), significantly more patients with EOCRC received adjuvant chemotherapy (71% vs. 39%, respectively; *p* < 0.001) and adjuvant oxaliplatin (81% vs. 50%, respectively; *p* < 0.001).

After a median follow-up of 25 months, 613 patients died. The median OS was not reached in either group (*p* = 0.270). Three-year OS was 80% and 67% in the younger and older groups, respectively (Figure 1). In patients with localized disease who underwent surgery (*n* = 1161), 133 events for time to recurrence (TTR) were observed. The median TTR was not reached in either group (*p* = 0.87). Three-year DFS was 86% and 73% in the younger and older groups, respectively (Figure 2). In patients with metastatic disease (de novo or metachronous; *n* = 449), significant differences were found between each group’s median OS, which was 31.4 months (95% CI: 18.9–43.9) in the EOCRC group and 18.2 (95% CI: 14.5–21.9; *p* = 0.049) in the older group (Figure 3). In the subgroup of patients with RAS/BRAF wild-type metastatic CRC, no differences in OS were found between the EOCRC and the older groups (*p* = 0.401). In patients with RAS/BRAF-mutated metastatic CRC, the median OS was significantly longer among patients with EOCRC than among patients aged over 50 years old (NR vs. 19.5 months (95% CI, 15.1–23.9), with the HR for death being 0.3 (95% CI, 0.1–0.8; *p* = 0.025) (Figure 4).

Regarding the impacts of COVID-19 on the staging and pathological features of EOCRC patients at diagnosis, 55 patients were diagnosed during the year before the initiation of the COVID-19 lockdown (14 March 2020), and 13 patients were diagnosed during the 365 days after that date. The main features identified at EOCRC diagnosis are depicted in Table 2. In total, 61% of patients in the post-pandemic group were diagnosed with metastatic disease, while only 29% had stage IV cancer at diagnosis in the pre-pandemic subgroup (*p* = 0.028). Moreover, 75% of patients diagnosed in the second semester post-pandemic were metastatic at presentation (Figure 5). High tumor budding, involved surgical margins, lymphovascular and perineural invasion were also pathological features more frequently observed in the post-pandemic subgroup, although the results were not statistically significant.

After a median follow-up of 34 months (39 and 17 months in the pre- and post-pandemic groups, respectively), 19 deaths were observed. Patients in the post-pandemic group had a significantly worse median OS than those in the pre-pandemic group (NR vs. 63 months (95% CI not estimated), respectively; *p* = 0.016) with an HR for death of 4.2 (95% CI: 1.1–15.8) At 12 months, 95% and 75% of patients were alive in the pre- and post-pandemic groups, respectively (Figure 6).

## 4. Discussion

In this study, we assessed the clinico-pathological features and outcomes of patients with EOCRC in our area. We found that 5% of our patients were diagnosed with CRC at a young age, and they presented with more advanced and histologically aggressive features than their older counterparts. Also, a more aggressive treatment strategy was found in patients with EOCRC. However, this context did not translate into poorer survival outcomes. Regarding the impacts of COVID-19 pandemic on this vulnerable subgroup of patients, we did not find significant differences in staging and aggressive features at presentation, though an alarming rise in metastatic patients at diagnosis was observed and should not be overlooked.

A lower overall percentage of patients are considered to have EOCRC in this cohort (5%) compared to other series [3], although this fact might be influenced by a general older population living in the area associated with our institution. In our cohort, patients with EOCRC were diagnosed at a more advanced stage (T, N and M) and with features of worse prognosis, such as perineural invasion, signet ring cells and higher grade. This observation might be the consequence of a more aggressive intrinsic tumor biology, a delay in the diagnosis and the absence of systematic screening in patients under 50 years. Our findings were also identified in previous studies [19]. The prevalence of stage III–IV disease at diagnosis was reported to be significantly higher in EOCRC patients in large population-based cohorts (53–72% in EOCRC vs. 41–63% in later-onset CRC across different studies [16,17,18]. The presence of signet ring cells, which is a known bad prognosis factor, is most frequent in patients younger than 30 years old, accounting for 6.3% of CRC patients, compared to 1–2% in CRC patients older than 30 years old [20]. Among metastatic CRC patients, Willauer et al. [20] described a prevalence of 6% of MSI-H in EOCRC vs. 1.6% in later-onset patients. Considering all EOCRC patients independent of stage at diagnosis, Antelo et al. reported a 23% prevalence of MSI-H [21], compared to an estimate of 15% of CRC patients overall [22]. In the largest cohort of CRC patients (>36,000 patients) reported to date [20], no significant differences were found regarding KRAS and NRAS mutations, although EOCRC patients were significatively less likely to present BRAF V600E mutations than later-onset CRC patients (5% vs. 10%; *p* < 0.001). Interestingly, the rate of obesity in EOCRC patients was significantly higher, which is a known risk factor for CRC. Contrary to previously published studies, no differences regarding the location of the primary tumor were found in this population of CRC patients. Patients with EOCRC were more aggressively treated than their older counterparts. This difference is remarkable in the adjuvant setting, in which significantly more patients with EOCRC received adjuvant treatment, and more of those patients were administered oxaliplatin combinations. Previous reports stated that patients with EOCRC are more two-to-four times more likely to receive adjuvant chemotherapy and multiagent regimens than older patients, adjusting for significant age-related variations in disease stage and treatment administration [2]. Although patient willingness, physician attitudes, or the belief that young adults will tolerate more aggressive treatments may have influenced these practices, they did not significantly impact overall survival [23].

The prognosis and survival of patients with EOCRC compared to older patients is inconsistent [15,23,24,25,26]. No differences in outcomes in the whole population and patients with localized disease were found. Other authors also found that although patients with localized EOCRC received more aggressive surgeries and adjuvant treatments, these treatments did not have an impact in survival [21]. In the metastatic cohort, a significantly better OS in EOCRC was identified. These results might be explained based on differences found regarding biological features favoring a better prognosis for EOCRC, such as the absence of BRAF mutation and more frequent RAS/BRAF wild-type status, as well as the fact that younger patients tend to tolerate higher doses and more lines of chemotherapy. Interestingly, the prognosis of a patient according to their mutational profile is different between the younger and older cohorts. While we do not observe significant differences in overall survival in the RAF/BRAF wild-type population, patients with RAS/BRAF-mutated metastatic EOCRC perform significantly better than those in the RAS/BRAF-mutated older counterpart (Figure 4b). This result might be the consequence of biological characteristics not measured in the clinical practice in RAS/BRAF wild-type patients, such as APC mutations, which confer worse prognoses more frequently found in EOCRC [27]. Other alterations are known to be bad prognostic factors in the RAS/BRAF wild type, but to our knowledge, they have not been described as being more prevalent in EOCRC patients, such as PIK3CA mutations (a known negative predictor of response to EGFR inhibitors in RAS wild-type tumors [28]) and HER2 status (its expression in the membrane of the tumor cells is associated with a shorter progression-free survival (PFS) to EGFR inhibitors) [29]. Whether ultraselection of RAS/BRAF wild-type patients is the key to improving prognosis in this population is yet to be determined [30,31]. Surely, this underlines the importance of a performing a broader study of molecular alterations, such as next-generation sequencing in CRC patients and, more importantly, EOCRC patients, to better select systemic treatment in the metastatic setting.

The COVID-19 pandemic has directly impacted the diagnostic flow of patients with CRC, as screening colonoscopies were reduced at multiple institutions. In fact, a recent study showed that CRC screening decreased by 28–100% in several countries after the onset of the COVID-19 pandemic [32]. Consequently, a reduction of 29% in newly diagnosed CRC cases was observed compared to 2019. New CRC patients in 2020 were less likely to be diagnosed with early-stage (stages I–III) CRC (63% vs. 78%) [33]. Patients with EOCRC are not usually included in CRC screening programs. A recent systematic review of 39 studies found decreases of −46%, −44% and −51% in breast, CRC and cervical cancer screenings, respectively, during the pandemic, which could result in an increase in avoidable cancer deaths [34]. A recent report from a Spanish institution did not find significant differences in the staging of the EOCRC at diagnosis before and after the COVID-19 pandemic, as well as in days to treatment initiation or enrollment in clinical trials, although an increase in EOCRC diagnosis was observed [35]. We report an alarming increase in metastatic disease at diagnosis after the COVID-19 pandemic, especially in the second semester after the lockdown (71% of patients), which probably impacted the median OS of patients with EOCRC diagnosed within the 365 days following the start date of the Spanish COVID-19 lockdown. Other aggressive clinico-pathological features were more frequently observed in the post-pandemic group; however, it did not reach statistical significance. Patients with EOCRC are an especially vulnerable population as the time to diagnosis from onset of symptoms is already high. In fact, more than 60% of patients with EOCRC waited more than three months since noticing symptoms to visit a doctor. In addition, during the COVID-19 pandemic, the fear of being infected led to minimized human interaction, including face-to face healthcare consultations [36]. The collapse of primary care and the healthcare system, the delay in diagnostic procedures and other factors may have directly impacted EOCRC diagnoses. Nevertheless, longer follow-up is needed to assess the long-term consequences of the COVID-19 with regard to this vulnerable population.

The main limitations to our study include the small sample size and its retrospective and unicentric nature. The former limitation mainly affects the way in which the COVID-19 pandemic impacted patients with EOCRC because although we included, in total, 1475 patients, only 68 patients were included in the COVID-19 subanalysis. Furthermore, although the differences in median OS between the pre- and post-pandemic groups could be explained based on the baseline characteristics of the patients, the follow-up of the post-pandemic group remains undetermined. Overall, despite the significant results, we must be careful when drawing conclusions. Although we seem to be recovering from the COVID-19 pandemic, protocols and recommendations are needed to overcome outbreaks or similar circumstances in the future. In an effort to harmonize the management of cancer patients during the COVID-19 pandemic, international societies, such as the European Society for Medical Oncology (ESMO), published their own recommendations for the management of patients with cancer during the pandemic [37]. Also, an online resource grading priorities regarding the diagnosis or treatment of CRC patients as high, medium or low during the pandemic is available at ESMO.org (URL: https://www.esmo.org/guidelines/cancer-patient-management-during-the-covid-19-pandemic/gastrointestinal-cancers-colorectal-cancer-crc-in-the-covid-19-era; accessed on 1 June 2023).

## 5. Conclusions

The rising incidence of EOCRC is a major concern among oncologists. These patients’ diseases are diagnosed at a more advanced stage and have aggressive features. Although median OS is longer among patients with metastatic EOCRC, the COVID-19 pandemic could have influenced the diagnosis and survival of this particularly vulnerable population. The long-term consequences of COVID-19 remain underexplored.

## Figures and Tables

**Figure 1 cancers-15-04242-f001:**
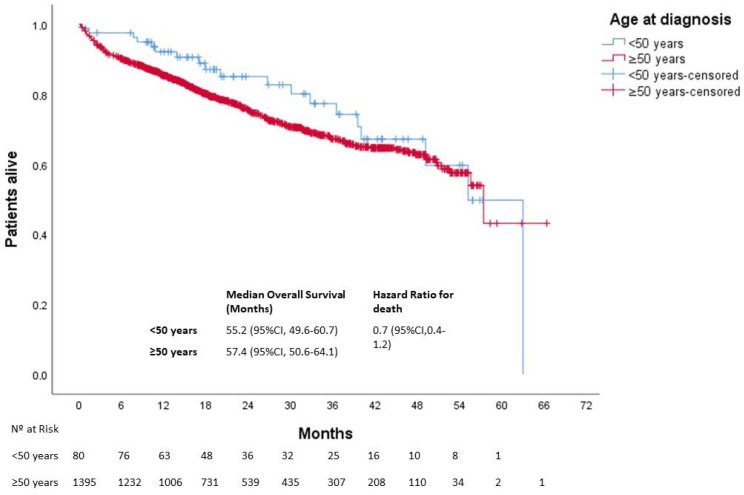
Overall survival of patients with EOCRC according to age at diagnosis. CI, confidence interval.

**Figure 2 cancers-15-04242-f002:**
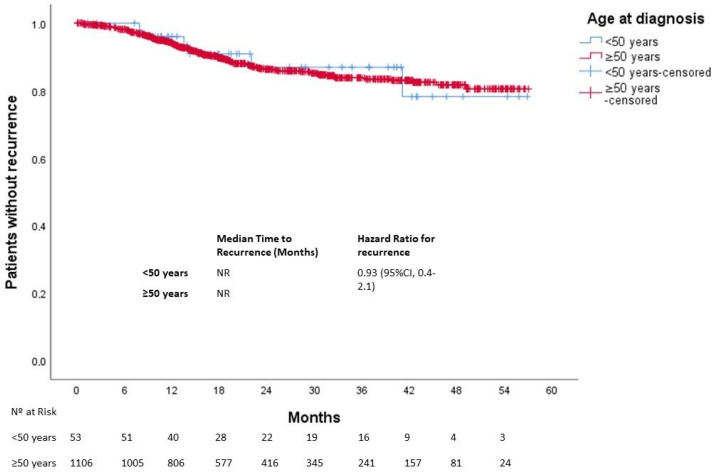
Time to recurrence in patients with localized disease according to age at diagnosis.

**Figure 3 cancers-15-04242-f003:**
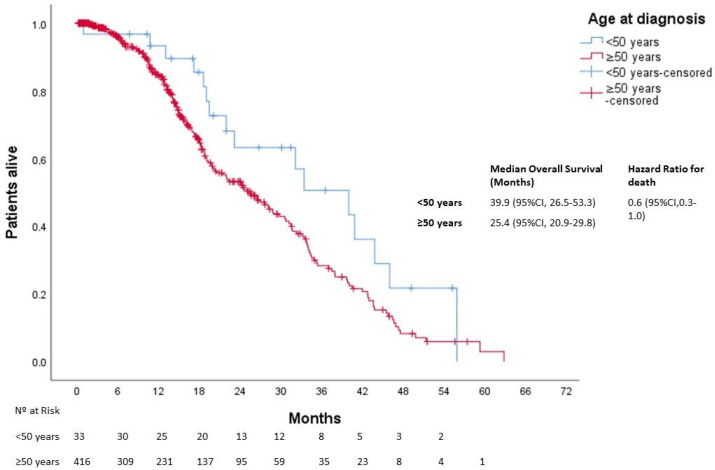
Overall survival in patients with metastatic disease according to age at diagnosis.

**Figure 4 cancers-15-04242-f004:**
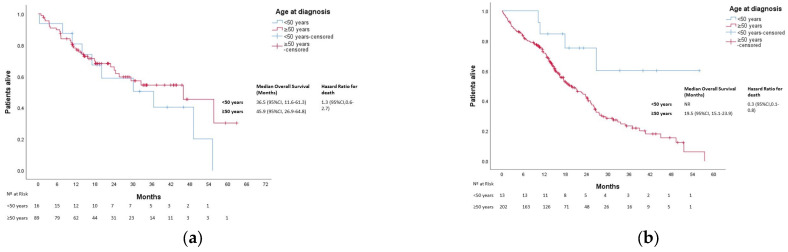
Overall survival in patients with metastatic CRC according to age and mutational profile. (**a**) Overall survival in patients with RAS/BRAF wild-type metastatic CRC according to age at diagnosis. (**b**), Overall survival in patients with RAS/BRAF-mutated metastatic CRC according to age at diagnosis. NR, not reached.

**Figure 5 cancers-15-04242-f005:**
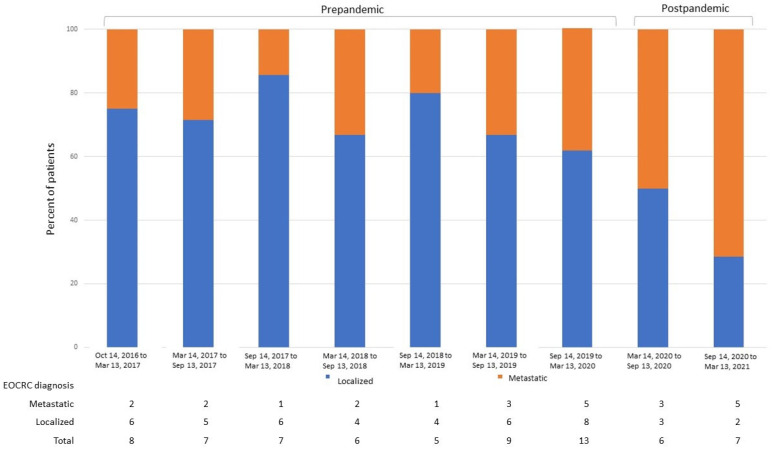
Percent of patients with EOCRC diagnosed with localized and metastatic disease according to the date of diagnosis. EOCRC, early-onset colorectal cancer.

**Figure 6 cancers-15-04242-f006:**
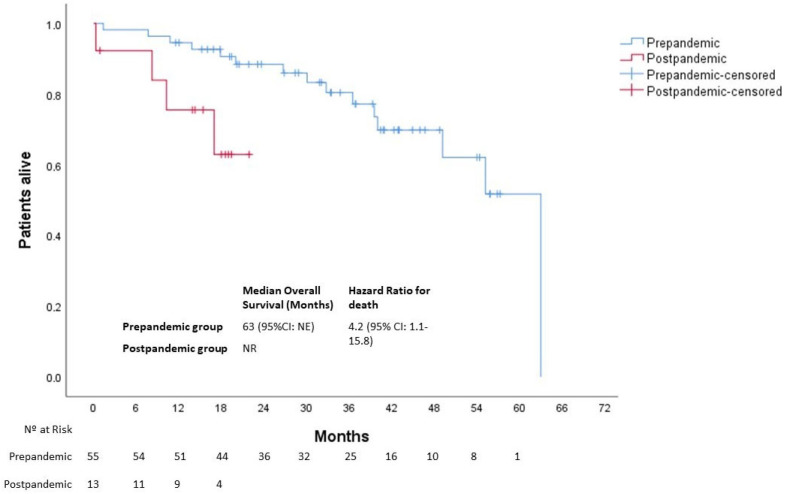
Overall survival of patients with EOCRC according to the date at diagnosis. NE, not estimated; NR, not reached.

**Table 1 cancers-15-04242-t001:** Main characteristics of the patients.

Characteristic	<50 Years Old (*n* = 80)	≥50 Years Old (*n* = 1395)	*p* Value
Sex (female), *n* (%)	33 (41.3%)	573 (41.1%)	0.975
Age, mean (SD)	44.9 (4.2%)	73 (10.5%)	0.001
ECOG PS 0-1 (compared to 2-4) *n* (%)	73 (93.6%)	1182 (87.8%)	0.122
BMI, *n* (%)			0.007
Underweight	3 (5%)	23 (2%)	
Healthy weight	29 (50%)	336 (36%)	
Overweight	11 (19%)	382 (41%)	
Obesity	15 (25%)	184 (19%)	
Primary tumor location, *n* (%)			0.34
Right colon	26 (32%)	474 (34%)	
Left colon	25 (31%)	518 (37%)	
Rectum	29 (36%)	432 (29%)	
T stage at diagnosis, *n* (%)			0.050
1	5 (7%)	180 (14%)	
2	5 (7%)	156 (12%)	
3	30 (42%)	522 (42%)	
4	31 (43%)	383 (30%)	
N stage at diagnosis, *n* (%)			0.001
0	22 (31%)	67 (53%)	
1	25 (35%)	335 (26%)	
2	24 (33%)	263 (20%)	
Stage, *n* (%)			0.002
I	7 (8%)	274 (19%)	
II	14 (3%)	372 (26%)	
III	32 (40%)	462 (33%)	
IV	27 (33%)	287 (20%)	
Stage IV at diagnosis, *n* (%)	27 (33%)	287 (20%)	0.005
Lymphovascular invasion, *n* (%) *	29 (46%)	420 (38%)	0.171
Perineural invasion, *n* (%) *	21 (34%)	246 (22%)	0.029
Budding, *n* (%) *			0.050
Low	17 (39%)	435 (54%)	
Medium–high	26 (60%)	371 (46%)	
Mucinous histology, *n* (%)	13 (18%)	156 (12%)	0.160
Signet ring cell, *n* (%)	5 (6%)	30 (2%)	0.020
High grade (G3), *n* (%)	13 (18%)	98 (8%)	0.004
Deficient mismatch repair, *n* (%)	11 (14%)	115 (9%)	0.120
Mutational profile, *n* (%)			0.020
BRAF	1 (2%)	75 (19%)	
KRAS	13 (35%)	158 (41%)	
NRAS	0 (0%)	20 (5%)	
Wild type	23 (62%)	131 (34%)	
Treatment:			
Surgery	65 (81%)	1174 (84%)	0.490
Chemotherapy	68 (85%)	872 (62%)	<0.001
Adjuvant chemotherapy	39 (71%)	437 (39%)	<0.001
Adjuvant oxaliplatin	30 (81%)	219 (50%)	<0.001

ECOG, Eastern Cooperative Oncology Group Performance Status; BMI, body mass index; * only performed in patients with resected localized disease.

**Table 2 cancers-15-04242-t002:** Characteristics of EOCRC patients before and after the COVID-19 pandemic.

	Pre-Pandemic Group (*n* = 55)	Post-Pandemic Group (*n* = 20)	*p* Value
T4, *n* (%) *	13 (34%)	3 (60%)	0.344
Lymph node positive, *n* (%) *	25 (65%)	3 (60%)	1.000
Metastasic disease at diagnosis, *n* (%)	16 (29%)	8 (61%)	0.028
Margins affected, *n* (%) *	4 (8%)	2 (28%)	0.17
Bowel obstruction at diagnosis *	4 (7%)	0 (0%)	1.000
Bowel perforation at diagnosis *	2 (3%)	0 (0%)	1.000
Lymphovascular invasion, *n* (%) *	19 (43%)	5 (71%)	0.232
Perineural invasion, *n* (%) *	14 (31%)	4 (57%)	0.226
High tumoral budding, *n* (%) *	15 (53%)	5 (83%)	0.364

* Only patients with resected localized disease at diagnosis were included.

## Data Availability

The data presented in this study are available on request from the corresponding author.

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
