# Peer review of "Clinico-Pathological Features, Outcomes and Impacts of COVID-19 Pandemic on Patients with Early-Onset Colorectal Cancer: A Single-Institution Experience"

_cancers, 2023, doi:10.3390/cancers15174242_

Round 1
Reviewer 1 Report
Overview: This is a single institutional study of 1475 CRC patients, of which 5.4% were classified as early onset (diagnosed <50yo). Clinicopathologic features were noted, in comparison to average-onset and pre/post COVID.
Specific comments:
Major points:
1. I did not see any mention of the possible role of antibiotics use and EO-CRC.
2. The OS curves are misleading as stage IV was group with non-metastatic. Given that it was known that there was stage imbalance between the groups, the KM curves should be by stage and then by age or not at all as the overall group does not signify an age-specific phenomenon.
3. It would be preferred if Figure 5 could also show total numbers of cases, not just % to give a sense of the ricing incidence as well as breakdown by stage
4. Is there any information on treatment? That would greatly enrich this analysis. Information on presenting symptoms and differences between the group would also add to the analysis.
Minor points:
1. Intro should also note disparities by ethnic group, not just age of onset.
This is fine
Author Response
Thank you for taking your time to review the manuscript. I modified the manuscript accordingly and hopefully improved it.
- I did not see any mention of the possible role of antibiotics use and EO-CRC.
- In line 77 we mentioned the antibiotic use as a possible risk factor for EOCRC. It is definitely an interesting topic, however, as we do not assess the role of antibiotics in our work (due to the retrospective nature of the study) I think it should not be explained in more detail.
- The OS curves are misleading as stage IV was group with non-metastatic. Given that it was known that there was stage imbalance between the groups, the KM curves should be by stage and then by age or not at all as the overall group does not signify an age-specific phenomenon.
- Figure 1 is the overall survival in the whole population according to the age at diagnosis. As, as you mentioned, there is an imbalance between groups in the stage at diagnosis, figure 2 refers to the time to recurrence in the localized setting and figure 3 refers to the overall survival in the metastatic setting according to age.
- It would be preferred if Figure 5 could also show total numbers of cases, not just % to give a sense of the ricing incidence as well as breakdown by stage
- Thank you for the suggestion. I added to the figure the number of cases en each bar and the total number of cases in each time frame.
- Is there any information on treatment? That would greatly enrich this analysis. Information on presenting symptoms and differences between the group would also add to the analysis.
- Thank you for the suggestions. We added information about the treatment. In fact, we found significant differences according to the use of chemotherapy. Due to its retrospective nature, in significant amount of patient’s data regarding the presenting symptoms and times are missing or are not reliable, so we decided to omit it.
Minor points:
- Intro should also note disparities by ethnic group, not just age of onset.
- Thank you for the suggestion. I added a sentence in the introduction about the disparities by ethnic group.
Reviewer 2 Report
Thank you for the opportunity to review this manuscript. My comments are as follows:
1. The findings that EOCRC are biologically more aggressive and present at more advanced stages are not new.
2. The authors mention in the manuscript several times regarding the rising incidence of EOCRC. Could the authors show the year-on-year incidence of EOCRC over the study period?
3. Is there a typo error in line 251 – “has decreased from 28 to 100% in different countries”
4. There has been some emphasis in this manuscript on the increase in proportion of metastatic disease amongst EOCRC post pandemic. The absolute number is very small (8/20) and authors have rightly mentioned this as an important limitation.
5. Nonetheless, as this is highlighted as one of the key findings of this study, the authors should postulate why this increase was observed and the influence of the pandemic/ lockdown protocols. Were younger patients found to have ignored symptoms or postponed their appointments because of the pandemic? Or where their specific hospital policies that prioritized timely investigations for older patients?
6. I think this study can be made more interesting if the authors can give further insight into the impact of the pandemic on younger patients seeking a diagnosis within the Spanish context and brief recommendations for similar circumstances in the future.
Quality of English is good
Author Response
- The findings that EOCRC are biologically more aggressive and present at more advanced stages are not new.
- I agree. Probably the most important part of the study was the impact of COVID-19 on EOCRC diagnosis, and how important is to maintain an adequate healthcare structure.
- The authors mention in the manuscript several times regarding the rising incidence of EOCRC. Could the authors show the year-on-year incidence of EOCRC over the study period?
- I agree it is an interesting data. However, only data from 2017-to 2021 are available, and COVID-19 may have impacted in EOCRC diagnosis. Nevertheless, following your suggestion and the one from the other reviewer, I added in figure 5 the absolute numbers of patients diagnosed per 6 months period.
- Is there a typo error in line 251 – “has decreased from 28 to 100% in different countries”
- Thank you, modified accordingly.
- There has been some emphasis in this manuscript on the increase in proportion of metastatic disease amongst EOCRC post pandemic. The absolute number is very small (8/20) and authors have rightly mentioned this as an important limitation.
- Agree
- Nonetheless, as this is highlighted as one of the key findings of this study, the authors should postulate why this increase was observed and the influence of the pandemic/ lockdown protocols. Were younger patients found to have ignored symptoms or postponed their appointments because of the pandemic? Or where their specific hospital policies that prioritized timely investigations for older patients?
- Thank you for the suggestions, I guess younger patients, who already ignore suggestive symptoms postponed their appointments. I added a paragraph in the discussion.
- I think this study can be made more interesting if the authors can give further insight into the impact of the pandemic on younger patients seeking a diagnosis within the Spanish context and brief recommendations for similar circumstances in the future.
- Thank you for the suggestion. Although the suggestion is very appealing, I think that official recommendation should be made at a national-worldwide level after all the consequences of covid-19 are made. So, it is not us to do such recommendations. We included a paragraph mentioning that international societies have made recommendations on the management of patients with covid that can be applied to future pandemics.
Round 2
Reviewer 1 Report
The manuscript is much improved.
1. Figure 5 - y-axis missing as well as a description of what the colors mean. I think the total should be absolute not % so it is less misleading
2. There needs to be more information about what treatment data is and is not available. This should also be discussed more significantly in the limitations, and the small numbers of patients with EOCRC in this sample set should be noted
3. This conclusion sentence (Although median OS is longer among patients with metastatic EOCRC, COVID-19 had a huge impact on the diagnosis and survival of this especially vulnerable population.) is not well supported by the paper due to the limitations in treatment information and ability to adjust for other key factors.
4. Line 276: correct this - including fate-to face
Some correction is needed, but not major
Author Response
- Figure 5 - y-axis missing as well as a description of what the colors mean. I think the total should be absolute not % so it is less misleading.
- Thank you for the suggestion. I have added the y-axis description. The color description is already in the figure. Although the absolute number would provide more raw information, I think that the key of this figure is to show that the percent of patients with metastatic disease has increased.
- There needs to be more information about what treatment data is and is not available. This should also be discussed more significantly in the limitations, and the small numbers of patients with EOCRC in this sample set should be noted.
- Thank you for the suggestion. Further details in the treatment received were added to the results, and the discussion was also enlarged. Agree that the main limitation is the small sample size and thus it is mentioned in the limitations.
- This conclusion sentence (Although median OS is longer among patients with metastatic EOCRC, COVID-19 had a huge impact on the diagnosis and survival of this especially vulnerable population.) is not well supported by the paper due to the limitations in treatment information and ability to adjust for other key factors.
- Modified to a less vehement conclusions
- Line 276: correct this - including fate-to face
- Corrected. Thank you.